# The impact of digital inclusive finance on environmental pollution: A case study of air pollution

Zexing Wang[1], Min Fan[2]*, Yaojun Fan[3]

1 UCSI Graduate Business School, University College Sedaya International, Kuala Lumpur, Malaysia,
2 School of Economics, Lanzhou University, Lanzhou, China, 3 Chinese International College, Dhurakij
Pundit University, Bangkok, Thailand

* 1160529881@qq.com

**Data Availability Statement:** This research conducted empirical analysis based on a dataset covering 265 cities in China from 2005 to 2021. The PM2.5 concentration data were compiled from the data of the Atmospheric Composition Analysis

## Abstract

This study delves into the impact of digital inclusive finance on environmental pollution, with a specific focus on air pollution. Utilizing data from 265 Chinese cities, advanced econometric methods such as the bi-directional fixed effects model, threshold model, spatial Durbin model, and multi-period difference-in-differences model are employed, incorporating a variety of control variables. The empirical findings indicate that digital inclusive finance significantly reduces air pollution. This mechanism chiefly operates through enhancing public environmental consciousness and fostering green technological innovation. The study also uncovers the spatial spillover effect and non-linear characteristics of digital inclusive finance on air pollution, along with its interactive effects with specific policies (e.g., smart city pilot policies and the "major protection, no major development" initiative). Moreover, heterogeneity analysis reveals regional variations in the environmental effects of digital inclusive finance. These insights provide a novel perspective on the relationship between financial technology and environmental protection and offer crucial guidance for policymaking.

## 1 Introduction

The dawn of the 21st century witnessed a rapid rise in digital inclusive finance globally, especially in developing countries, propelled by the swift advancement of internet and mobile communication technologies. This financial service filled gaps left by traditional banking systems, notably in areas underserved by banks. The widespread accessibility of smartphones and the internet has enabled more people to engage in financial transactions digitally, marking not just technological progress but a significant innovation in financial models. Digital inclusive finance, through simplifying procedures and reducing transaction costs, has made services like micro-loans, insurance, and investments more accessible to low-income individuals and small and micro enterprises. Its profound impact on society and the economy includes providing critical opportunities for groups traditionally excluded from financial services, enhancing their economic activities and life quality. It has also fostered gender equality, particularly by empowering women's economic independence and financial autonomy [1]. At the macroeconomic

Group, the digital inclusive finance data were from the "Peking University Digital Inclusive Finance Index (2011-2021)" published by the Peking University Digital Finance Research Center (https://idf.pku.edu.cn/), the public environmental awareness data were from the Baidu Index, the green patent application data were from the China National Intellectual Property Administration, and other city-level data were from the China City Statistical Yearbook. Since these data are primarily from publicly accessible third-party sources, we are unable to provide the dataset directly. For interested researchers, the following are instructions for accessing the third-party data: City-level PM2.5 concentration data: Obtained from the Atmospheric Composition Analysis Group (http://fizz.phys.dal.ca/~atmos/martin/?page_id=140). Digital inclusive finance data: From the "Peking University Digital Inclusive Finance Index (2011-2021)" published by the Peking University Digital Finance Research Center (https://idf.pku.edu.cn/). Public environmental awareness data: Obtained from the Baidu Index (https://index.baidu.com/). Green patent application data: Obtained from the China National Intellectual Property Administration (https://www.cnipa.gov.cn/). Other city-level data: Can be accessed through the China City Statistical Yearbook, which is available on the website of the National Bureau of Statistics (https://www.stats.gov.cn/). Since the data are from third-party sources, our research team is unable to provide direct data download links or datasets. During this research, we have ensured that all data used can be accessed by other researchers through regular channels, without the use of any special access privileges.

**Funding:** The author(s) received no specific funding for this work.

**Competing interests:** The authors have declared that no competing interests exist.

level, it supports small and micro businesses and entrepreneurs, stimulating job growth, diversifying national economies, and alleviating poverty [2]. Additionally, digital inclusive finance enhances the efficiency of the financial system and reduces transaction costs [3]. Globally, it signifies a shift in the economic structure, especially impacting developing countries, which have leapfrogged into the digital finance era, transforming their financial landscapes and bringing new dynamics to the global financial market. However, as digital inclusive finance evolves, challenges in data security, privacy protection, and regulation become increasingly prominent, affecting not only the safety and stability of financial services but also indirectly influencing the formulation and implementation of environmental policies [4, 5].

Currently, environmental pollution represents one of the most severe challenges faced globally. The acceleration of industrialization and urbanization has intensified various forms of pollution worldwide. Air pollution, particularly from industrial emissions, vehicle exhaust, and energy production (especially the combustion of fossil fuels), releases substantial greenhouse gases and toxic substances like sulfur dioxide and nitrogen oxides, severely impacting air quality and posing direct threats to human health [6]. Urban areas, with their traffic congestion and vehicle emissions, face pronounced air pollution issues due to high population density and concentrated motor vehicles [7, 8]. Water pollution is also a global environmental concern, stemming from industrial wastewater discharge, agricultural chemical runoff, and improper treatment of domestic sewage. These pollutants not only disrupt ecosystems in rivers, lakes, and oceans but also affect human access to clean water sources [9].

In the current context of increasingly severe global climate change and environmental degradation, finding effective solutions has become an urgent global task. Air pollution, as one of the main environmental issues affecting the health and quality of life of billions of people globally, urgently requires innovative strategies and technologies for its management. Against this backdrop, digital inclusive finance, as an emerging financial service model characterized by its widespread coverage and low cost, is considered a potential tool for promoting environmental sustainability. For instance, it can alleviate environmental pollution by facilitating green finance and sustainable investments, but at the same time, it could also exacerbate environmental pressures due to the extensive use of technological equipment and energy consumption [10]. However, despite significant progress in improving financial inclusivity, the role and potential of digital inclusive finance in environmental protection have not been fully explored and applied. Hence, understanding the complex relationship between digital inclusive finance and environmental pollution is crucial for formulating effective policies and practices to achieve sustainable development. This understanding not only contributes to solving current environmental issues but is also key to the sustainable development of the global economy.

This study aims to explore and validate how digital inclusive finance impacts air pollution, with particular attention to its role in raising public environmental awareness, promoting green technological innovation, and achieving policy spillover effects. Through a multidimensional analysis of the relationship between digital inclusive finance and air pollution, this paper seeks to reveal its environmental effects under different geographical and economic backgrounds, offering scientific evidence and empirical support for the formulation of related policies.

## 2 Literature review and theoretical analysis

### 2.1 Literature review

In recent years, with the digital transformation of the global economy, digital inclusive finance has emerged as a significant trend. This financial model, leveraging digital technologies, particularly the internet and mobile communication, has notably improved the financial accessibility

for low-income and marginalized groups, thereby fostering inclusive economic growth. Studies have shown that the development of digital inclusive finance has had substantial impacts on the economy in several aspects. Primarily, it has played a crucial role in enhancing financial inclusiveness. Research by Yang et al. (2020) [11] found that in China, digital financial inclusivity significantly increased the per capita disposable income of urban and rural residents, spurring economic growth and entrepreneurial activities. Additionally, Yang & Zhang (2020) [12] noted that digital inclusive finance aids the sustainable growth of small and micro enterprises, especially in private and high-tech sectors. The influence of digital inclusive finance on economic growth and innovative development is also significant. Zuo (2021) [3] observed that digital inclusive finance positively impacts economic growth and significantly narrows the income gap between urban and rural areas. Ji et al. (2021) [13] further emphasized the role of digital inclusive finance in fostering entrepreneurial initiatives and financial participation among residents. Research by Wang et al. (2023) [14] showed that digital inclusive finance directly positively affects economic growth and innovation and indirectly influences them by improving individual disposable income and education levels. Lastly, Liu et al. (2022a) [15] demonstrated that digital inclusive finance had a significant positive effect on the economic development of Shandong Province, China, a trend corroborated in China's other 31 provinces and cities. These studies indicate that digital inclusive finance plays a vital role not only in promoting economic inclusiveness but also positively affects overall economic development, innovation, and regional economic equilibrium.

Against the backdrop of the global economy's digital transformation, digital inclusive finance, as an emerging financial model, has rapidly developed worldwide, drawing widespread attention to its environmental impacts. The positive influence of digital inclusive finance on environmental sustainability primarily manifests in supporting environmentally friendly economic activities and investments [16]. For instance, research by Tariq et al. (2022) [17] found a significant positive correlation between digital finance, financial inclusivity, and environmental sustainability in emerging Asian countries. Additionally, Zheng et al. (2021) [18] pointed out that inclusive finance can promote corporate environmental investments, alleviate credit constraints, and enhance technological innovation in environmental protection. This suggests that digital inclusive finance has the potential to foster green investments and environmental projects. However, the development of digital inclusive finance also presents environmental challenges. Moriarty & Honnery (2016) [19] noted in their study that increased digitalization might lead to higher energy demand, thus exacerbating environmental pressures. Furthermore, research by Wang et al. (2022a) [5] discovered that while the development of the digital economy reduced pollution in urban areas, it increased emissions in non-urban regions.

Despite the abundant literature on digital inclusive finance and its impact on economic inclusivity, studies linking it to environmental pollution remain relatively limited. Specifically, regarding how digital inclusive finance affects specific environmental issues, such as air pollution, water pollution, and soil degradation, there is a scarcity of research. This paper aims to explore the specific connection between digital inclusive finance and environmental pollution, particularly its impact on air pollution, to fill this gap in the literature and provide guidance for future research and policymaking.

## 2.2 Analysis of the impact of digital inclusive finance on air pollution

**2.2.1 Theoretical framework.** The impact of digital inclusive finance on environmental pollution, especially air pollution, can be analyzed from the perspectives of Environmental Economics and Social Capital Theory. The theory of Environmental Economics emphasizes

that the proliferation and innovation of financial services can directly affect the use and protection of environmental resources by changing investment and consumption patterns [20, 21]. Digital inclusive finance, by lowering the barriers to investing in clean technologies, encourages capital to flow towards more environmentally friendly projects, which aligns with the concept of "market failure correction" in Environmental Economics [22].

Social Capital Theory provides a framework for understanding how digital inclusive finance can promote environmentally friendly behaviors by enhancing connections between communities and individuals. Putnam (1995) [23] suggested that the strengthening of social capital can improve the efficiency of public goods management, including environmental quality. As a tool for enhancing social capital, digital inclusive finance enhances financial inclusivity, enabling more people, particularly those in regions underserved by traditional financial systems, to participate in environmental projects [24].

Innovation Diffusion Theory also supports understanding how digital inclusive finance can promote the widespread acceptance and application of environmental technologies. Rogers (2003) [25] posited that the diffusion of technological innovations is a sociological process dependent on the technology's relative advantage, compatibility, complexity, testability, and observability. Digital inclusive finance, by providing necessary financial support and platforms, has accelerated the adoption of clean technologies such as solar and wind energy, not only improving the economic accessibility of these technologies but also the public's observability of their benefits.

In sum, through the market correction concepts of Environmental Economics, the community participation drive of Social Capital, and the technology promotion mechanisms of Innovation Diffusion Theory, we can deeply understand the positive impact of digital inclusive finance on air quality. These theoretical frameworks not only provide a solid theoretical foundation for our research hypothesis but also guide us in further exploring specific impact mechanisms.

**2.2.2 Economic behavior transformation.** Digital inclusive finance has a significant positive impact on air pollution by changing economic behavior. One of its core characteristics is the use of technological innovation to make financial services more widespread and convenient, thereby lowering barriers to investing in clean energy and environmental technologies. The proliferation of this financial model directly encourages individuals and enterprises to invest more easily in energy-saving and emission-reduction technologies, such as wind, solar, and other renewable energy projects. These projects reduce dependence on fossil fuels, significantly alleviating air pollution caused by traditional energy sources such as coal and oil [18, 26]. Additionally, digital inclusive finance plays a key role in promoting a green transformation of consumption patterns. By providing loans and other financial products, it motivates consumers and businesses to choose more energy-efficient and environmentally friendly products and services, such as the widespread adoption of electric vehicles and energy-saving appliances, which significantly reduce emissions of air pollutants [27, 28]. Enterprises also reduce energy use and pollution by adopting more efficient production technologies and practices, further diminishing their negative impact on air quality. Digital inclusive finance also accelerates the economic transition to eco-friendly and low-carbon models by promoting capital flows towards green and sustainable projects [29]. With more capital invested in these projects, clean technology development and application are spurred, generating new employment opportunities and economic growth points [30, 31]. This transformation is beneficial for long-term economic development and has profound implications for environmental protection and the reduction of air pollution [32, 33].

**2.2.3 Social structure improvement.** Digital inclusive finance also plays a significant role in social structure improvement. It provides key financial opportunities for regions

underserved by traditional financial services, enabling more people to participate in environmental projects [1, 34]. Such pervasive financial services are particularly helpful for communities that are overlooked by conventional finance due to geographical or economic reasons. For instance, rural residents can now access funding to invest in solar power generation and other renewable energy projects, improving local air quality and reducing reliance on coal and other polluting energy sources [35, 36]. Digital financial platforms, through educational content and interactive tools, play a crucial role in raising public awareness and education about environmental protection. These platforms not only provide basic knowledge about the environment but also showcase the specific effects of environmental behaviors, such as actual data and case studies on energy conservation and emission reduction, allowing the public to viscerally understand the positive impacts of their actions on the environment [18]. Moreover, financial inclusion promotes active participation in environmental projects by communities and small and micro-enterprises by providing them with the necessary financial support for implementing energy-saving renovations, developing environmental products, or improving waste management systems [37].

**2.2.4 Technology innovation and policy synergy.** Digital inclusive finance, as a driver of systemic change, also promotes the development of new environmental solutions. For example, digital technological innovation has led to the emergence of new tools such as smart energy management systems, which optimize energy use and effectively reduce wastage, thereby mitigating air pollution [38, 39]. Simultaneously, the growth of digital inclusive finance fosters the synergy between financial policies and environmental protection policies, allowing governments to more effectively implement tax and subsidy policies related to environmental protection, further advancing environmental protection and emission reduction projects [40, 41].

In conclusion, the following research hypothesis H1 is proposed:

H1: The development of digital inclusive finance contributes to reducing air pollution.

## 2.3 Digital inclusive finance, public environmental awareness, and air pollution

In today's world, environmental issues have become a global focus, and public attention to environmental protection is continuously increasing. This enhanced environmental awareness is crucial for addressing ecological challenges such as air pollution [42, 43]. In this context, digital inclusive finance, as an innovative financial mode, plays a key role in raising public awareness about the environment [44]. Through its extensive platform and technological advantages, digital inclusive finance not only provides financial services but also becomes an important channel for disseminating environmental knowledge and promoting environmental awareness [17]. Digital inclusive finance significantly contributes to raising public awareness about environmental protection, mainly through education and information dissemination provided by its technology platforms [45]. These digital platforms utilize interactive educational content, infographics, and online seminars, among other methods, to effectively disseminate knowledge about air pollution and environmental protection to the public. For example, they offer practical tips on energy saving and emissions reduction, advice on sustainable consumption, and the latest information on environmental regulations and policies, thus helping the public to deeply understand the causes and impacts of air pollution and methods to reduce pollution. Furthermore, by providing lending and investment options for green projects, digital inclusive finance encourages public direct involvement in environmental initiatives, enhancing their awareness of the importance of environmental protection [46]. The

widespread availability of this financial service not only allows individuals and enterprises to contribute to reducing air pollution but also enhances the public's overall understanding of environmental issues.

Therefore, through various efforts in education, participation, and awareness raising, digital inclusive finance plays an important role in promoting public attention to and involvement in environmental protection, positively impacting society's overall environmental consciousness and eco-friendly behaviors.

The enhancement of public environmental awareness plays a crucial role in reducing air pollution, manifested in changes in their behavior patterns and consumption habits [47, 48]. With increased awareness of environmental protection, consumers are increasingly inclined to choose eco-friendly products and services, such as opting for electric vehicles instead of gasoline cars and purchasing energy-efficient and low-emission products, directly reducing reliance on traditional energy sources and significantly reducing emissions of air pollutants [49, 50]. Additionally, the public is increasingly engaged in various environmental activities and initiatives, such as tree planting, community cleaning, and environmental advocacy, which not only improve community environmental quality but also help to enhance air quality [42]. Everyday energy-saving and emission-reduction measures adopted by the public, like conserving water and electricity, recycling, and waste sorting, further reduce environmental pollution [51]. This heightened environmental awareness also influences policy-making; as more people start paying attention to environmental issues, governments and policymakers are more likely to implement policies favoring environmental protection, such as raising emissions standards, supporting renewable energy, and reinforcing environmental protection regulations [52, 53]. The implementation of these policies further drives societal development toward a more environmentally friendly direction, effectively reducing air pollution [54]. Therefore, through active participation at the individual, community, and policy levels, the elevation of public environmental awareness plays a key role in significantly improving air quality and promoting sustainable environmental development.

In summary, the following research hypothesis H2 is proposed:

H2: Digital inclusive finance can alleviate air pollution by enhancing public environmental awareness.

## 2.4 Digital inclusive finance, green technology innovation, and air pollution

In the context of contemporary environmental protection and sustainable development, green technology innovation is widely regarded as a core strategy for reducing air pollution and addressing climate change [55]. These innovations span a broad spectrum from clean energy solutions to efficient pollution control technologies, aiming to foster a more eco-friendly and sustainable future. In this domain, digital inclusive finance, as an innovative financial model, plays a pivotal role in driving the development of these green technologies [56]. By providing more accessible funding and resources, digital inclusive finance significantly lowers the barriers to researching and commercializing green technologies, especially for small enterprises and startups [57]. These entities, often key sources of environmental innovation, now have access to rapid financing through digital inclusive finance platforms such as crowdfunding and microloans, accelerating the transformation from concept to market application [58]. Moreover, these platforms facilitate global information sharing and collaboration, providing efficient network platforms for technology transfer and knowledge sharing, speeding up the dissemination and application of environmental technologies [59]. Digital inclusive finance

also stimulates market demand for green technologies, creating favorable market conditions for their commercialization. As public environmental awareness grows, consumer demand for clean and sustainable products increases, driving businesses to invest in green technologies [60]. Additionally, these financial services influence the policy environment. As the role of green finance becomes more prominent in socio-economic contexts, policymakers are increasingly inclined to introduce policies supporting the innovation and application of environmental technologies, providing further support for green technologies [26]. Therefore, digital inclusive finance promotes green technology innovation and application by supporting funding, facilitating information and resource sharing, guiding market demand, and influencing the policy environment. This comprehensive support significantly contributes to the global goal of reducing air pollution and provides solid backing for achieving sustainable development goals.

In global efforts for environmental governance, green technology innovation plays a central role in reducing air pollution. These innovations include clean energy production, efficient energy use, and pollution control technologies [61, 62]. Particularly in the renewable energy sector, advancements in wind and solar technologies have significantly reduced dependency on fossil fuels, directly lowering greenhouse gas emissions and other harmful emissions from burning coal and oil [63, 64]. In the transportation sector, the development of electric vehicles and other low-emission transport solutions offers effective ways to reduce urban exhaust emissions, playing a key role in improving urban air quality with their zero-emission characteristics during operation [65, 66]. Additionally, the application of energy-saving technologies in buildings and industrial production indirectly reduces air pollution by increasing energy efficiency and optimizing resource use [67]. Similarly, technological innovations in clean production processes and the use of eco-friendly materials have significantly contributed to reducing the negative impact of industrial activities on air quality [68, 69]. Therefore, green technology innovation plays a decisive role in providing sustainable energy solutions, optimizing energy use, and reducing pollutant emissions, crucially supporting the goal of a clean and healthy environment.

In summary, the following research hypothesis H3 is proposed:

H3: Digital inclusive finance can alleviate air pollution by enhancing the level of green technology innovation.

## 2.5 Digital inclusive finance, spatial spillover effects, and air pollution

In the global effort to tackle air pollution, digital inclusive finance has not only produced positive effects in the areas it directly serves but also significantly benefitted the environmental quality of neighboring areas through spatial spillover effects [28, 70]. This effect highlights the indirect role of digital inclusive finance in promoting environmental protection and reducing air pollution, particularly in areas such as technology innovation dissemination, enhancing environmental awareness, and policy impact [10, 71]. First, the support of digital inclusive finance for green technology innovation is not limited to its immediate service areas. When enterprises or research institutions in one region successfully develop new environmental technologies with the help of digital inclusive finance, these technologies are likely to be adopted by neighboring regions through various channels [72, 73]. For example, clean energy projects supported by digital finance, such as efficient solar panels or wind energy technology, may attract interest and investment from surrounding areas, thus improving air quality in the entire region [74]. Second, the role of digital inclusive finance in enhancing public environmental awareness and knowledge sharing also has spatial spillover effects [75]. With the

proliferation of digital platforms, environmental initiatives and knowledge originating in one area can quickly spread to adjacent regions, sparking broader public involvement and eco-friendly actions. This sharing of knowledge and practices is crucial for enhancing environmental awareness and behavior across the region. Additionally, the influence of digital inclusive finance in shaping environmental policy-making has significant spatial spillover effects [76, 77]. Policymakers, inspired by observing successful environmental projects and policies driven by digital inclusive finance in neighboring areas, may adopt similar measures. This policy imitation and adoption not only enhance inter-regional cooperation and coordination in environmental protection but also accelerate the advancement and implementation of environmental policies throughout the region [78].

In conclusion, the development and application of digital inclusive finance have impacts on air pollution reduction that extend far beyond its direct service areas. Through spatial spillover effects, it influences a broader range of environmental quality. This effect underscores the important role of digital inclusive finance in global environmental governance, especially in promoting technology innovation dissemination, enhancing environmental consciousness, and influencing environmental policy. Therefore, digital inclusive finance is not only an innovation in the financial field but also a vital tool for advancing regional and global environmental protection. The following research hypothesis H4 is proposed:

H4: Digital inclusive finance also helps to reduce air pollution in neighboring areas.

## 2.6 Digital inclusive finance, non-linear effects, and air pollution

In the global challenge of addressing air pollution, the role of digital inclusive finance exhibits complex non-linear characteristics, evidenced by dynamic changes in its effectiveness in improving air quality [56, 72]. This non-linear effect reflects the complexity of the interaction between digital inclusive finance and environmental impacts, as well as its varying influence on environmental protection at different stages of development. This paper posits that in the early stages of digital inclusive finance development, its impact on reducing air pollution is most pronounced. During this phase, digital inclusive finance actively improves the environment by increasing green investments and promoting the development of environmental technologies [79]. For instance, by providing financial support for green energy projects, environmental innovation, and sustainable practices, digital inclusive finance effectively promotes the adoption of environmentally friendly technologies and practices during this period [56]. Additionally, environmental knowledge and practices are more widely disseminated through digital platforms, enhancing public awareness and participation in environmental protection [80].

As digital inclusive finance continues to develop and become more widespread, its marginal utility in environmental protection begins to diminish. This may be due to the widespread adoption of initial environmental technologies and practices reaching a saturation point in improving air quality. Additionally, increased energy demands accompanying the digitalization process, such as the energy consumption of data centers, may somewhat weaken the environmental benefits of digital inclusive finance [10]. At the same time, markets and policies may lag in addressing environmental issues, with new innovations and strategies requiring time to form and be implemented.

This non-linear effect emphasizes the need to fully consider the dynamism and complexity of digital inclusive finance's impact on the environment when formulating environmental protection strategies and financial policies. Understanding this non-linear relationship is crucial for optimizing environmental policies and ensuring coordinated development between

financial growth and environmental protection. Consequently, the following research hypothesis H5 is proposed:

H5: The impact of digital inclusive finance on air pollution exhibits non-linear effects.

## 3 Data and methodology

### 3.1 Model design

**3.1.1 Benchmark model.** As in previous studies, considering that individual and time factors may affect the regression results, this paper uses the research data of 265 cities in China to construct the following two-way fixed-effect model to examine the specific impact of digital financial inclusion on air pollution.

$$poll_{i,t} = \alpha_0 + \alpha_1 dfi_{i,t} + \delta X + \gamma_i + \omega_t + \varepsilon_{i,t} \tag{1}$$

Among them, poll represents the level of air pollution at the city level, dfi represents the level of digital financial inclusion in the city, and X represents the control variables, which are respectively economic development level (eco), population density (huma), science and technology expenditure (sci), foreign direct investment (out), industrial structure level (ind), and financial development level (fin). In addition, $\gamma_i$ and $\gamma_i$ represent the fixed effect of urban individuals and the fixed effect of time respectively.

**3.1.2 Mechanism test model.** In order to test the effect mechanism of digital inclusive finance on air pollution, and because the traditional intermediary model has major causal defects [81], the following improved model is used for reference to the practice of Jiang (2022b) [81] and Hu (2023) [82] to test the effect mechanism from two aspects: public environmental awareness and green technology innovation.

$$road_{i,t} = \alpha_0 + \alpha_1 dfi_{i,t} + \delta X + \gamma_i + \omega_t + \varepsilon_{i,t} \tag{2}$$

$$poll_{i,t} = \alpha_0 + \alpha_1 road_{i,t} + \delta X + \gamma_i + \omega_t + \varepsilon_{i,t} \tag{3}$$

In the given text, "road" represents a mechanism variable, which is specified by two different aspects: public environmental awareness and green technology innovation. Public environmental awareness is depicted using the Baidu Haze Index, which presumably tracks how frequently terms related to haze are searched for on the Baidu search engine, indicating the level of public concern about air quality issues. Green technology innovation is measured using the number of green patent applications, which reflects the extent of innovative activities in developing environmentally friendly technologies. As for other variables, they are defined consistently with Eq (1).

**3.1.3 Spatial spillover effect test model.** In addition, in order to investigate the spatial spillover effects of digital financial inclusion on air pollution and capture the economic links between regions and potential spillover effects, we introduced the following spatial Durbin model, the most common in spatial economic analysis. By introducing a spatial weight matrix, the model is able to capture and quantify economic linkages between regions and the resulting potential spillovers, thus providing a comprehensive framework to analyze the spatial impact of digital financial inclusion on air pollution. The common spatial geographic distance matrix is used in the analysis.

$$poll_{it} = \alpha_0 + \alpha_1 dfi_{it} + \beta_1 \sum_j w_{ij} dfi_{it} + \alpha_2 X + \beta_2 \sum_j w_{ij} X + \rho \sum_j w_{ij} poll_{it} + \gamma_i + \omega_t + \varepsilon_{it} \tag{4}$$

Where, $\rho$ represents the spatial lag autoregressive coefficient, $w_{ij}$ is the element of the spatial weight matrix.

**3.1.4 Nonlinear effect test model.** In order to further analyze the nonlinear characteristics of the impact of digital financial inclusion on air pollution, the panel threshold model was adopted for empirical test. This model choice is based on its advantages in dealing with nonlinear relationships in complex economic data, and is particularly suitable for exploring and verifying nonlinear relationships between variables. After threshold test, the dual threshold model is selected. The specific model is as follows.

$$poll_{i,t} = \alpha_0 + \delta_1 dfi_{it}*I(dfi \leq \gamma_1) + \delta_2 dfi_{it}*I(\gamma_1 < dfi \leq \gamma_2) + \delta_3 dfi_{it}*I(dfi > \gamma_2) + \delta X + \gamma_i + \omega_t + \varepsilon_{i,t} \quad (5)$$

Where I ($\cdot$) is the indicative function, $\gamma_1$, $\gamma_2$ are the threshold value, that is, when dfi is less than $\gamma_1$, the impact coefficient of digital inclusive finance on air pollution is $\delta_1$; when dfi is between $\gamma_1$ and $\gamma_2$, the impact coefficient of digital inclusive finance on air pollution is $\delta_2$; when dfi is greater, the impact coefficient of digital inclusive finance on air pollution is $\delta_3$.

## 3.2 Variable selection

**3.2.1 Dependent variable.** Air Pollution (poll): This study uses PM2.5 concentration data to measure the level of urban air pollution. PM2.5 has been a major air pollutant in China [10], especially since 2013, and is widely considered a key indicator for assessing air quality and the extent of air pollution. Due to its small particle size, which allows it to penetrate deeply into the lungs, it has a significant impact on human health and is a scientifically and universally accepted metric for measuring air pollution.

**3.2.2 Independent variable.** Digital Inclusive Finance (dfi): Drawing on the research by Fan & Chen (2022) [83], this article uses the Digital Inclusive Finance Index released by the Institute of Digital Finance at Peking University to measure the development level of digital inclusive finance in cities. The index is compiled by the Institute of Digital Finance at Peking University, which is a highly authoritative research institution in the field of digital finance in China. The construction of the Digital Inclusive Finance Index is based on extensive industry data, including but not limited to the ubiquity and utilization of digital payments, digital lending, digital insurance services, and the effectiveness of financial technology in improving inclusiveness. The method of compiling this index takes into account various factors such as the breadth of service, accessibility, cost of use, and user satisfaction, ensuring the comprehensiveness and depth of its measurement.

**3.2.3 Mechanism variables.** Public Environmental Awareness (car): Drawing from the research of Wu, Yang, & Sun (2022) [84], this paper uses the Baidu Haze Search Index to depict public environmental awareness. Internet-based data, reflecting public attention and behavioral intentions through online search behaviors, have become a novel indicator in the age of the internet. Baidu, as China's largest Chinese search engine, offers extensive coverage and high data accessibility. By analyzing search frequency and geographical location, the public's environmental concern in different regions of China can be effectively captured and compared. Thus, using the Baidu Haze Search Index provides a more direct understanding of the public's concern about air quality and environmental pollution.

Green Technology Innovation (gif): The study measures the level of urban green technology innovation using the number of green patent applications per 10,000 people. Patent applications are often seen as important indicators of technological progress and innovation capacity. In the field of green technology, the number of patent applications reflects a region's R&D efforts and innovative achievements in environmental and sustainable technologies. By measuring the number of green patent applications per 10,000 people, the research and innovation efforts and outcomes of cities in green technology can be understood, thus assessing their potential contribution to environmental improvement.

**Table 1. Variable definition.**

| | Variable name | Variable symbol | Measure of variable |
|---|---|---|---|
| Dependent Variable | Air Pollution | poll | PM2.5 concentration |
| Independent Variable | Digital Inclusive Finance | dfi | Digital Inclusive Finance Index |
| Mechanism Variables | Public Environmental Awareness | car | Baidu Haze Search Index |
| | Public Environmental Awareness | gif | The number of green patent applications per 10,000 people |
| Mechanism Variables | Economic Development Level | eco | Per capita GDP logarithm |
| | Population Density | huma | The logarithmic value of population per square kilometer |
| | Science and Technology Expenditure | sci | The proportion of technology expenditure to fiscal expenditure |
| | Foreign Direct Investment | out | The proportion of foreign direct investment in GPD |
| | Industrial Structure Level | ind | The proportion of output value of the tertiary industry to GDP |
| | Financial Development Level | fin | The proportion of loan balance from financial institutions to GDP |
| | Infrastructure Level | infra | Books in public libraries per 10,000 people |
| | Degree of government intervention | gov | Government spending as a share of GDP |

**3.2.4 Control variables.** The paper employs the level of economic development (eco), population density (huma), science and technology expenditure (sci), foreign direct investment (out), industrial structure (ind), level of financial development (fin), level of infrastructure (infra), and the degree of government intervention (gov) as the study's control variables. These variables enable a more comprehensive consideration of factors influencing air pollution, thereby allowing for a more accurate identification of the relationship between digital inclusive finance and air pollution. The variable definition table is shown in Table 1.

## 3.3 Data description

Considering the availability and continuity of urban data, this paper selects data from 265 Chinese cities as the basis for empirical research. The PM2.5 concentration data is derived from the Atmospheric Composition Analysis Group, digital inclusive finance data from the "Peking University Digital Inclusive Finance Index (2011–2021)," public environmental awareness data from the Baidu Index, green patent application data from the China National Intellectual Property Administration, and other city data from the "China City Statistical Yearbook." Additionally, some missing values are filled using linear interpolation. The descriptive statistics of the variables are presented in Table 2. The dependent variable poll (degree of air pollution) has

**Table 2. Descriptive statistics.**

| VarName | Obs | Mean | SD | Median | Min | Max |
|---|---|---|---|---|---|---|
| poll | 2915 | 0.412 | 0.150 | 0.383 | 0.116 | 0.868 |
| dfi | 2915 | 0.185 | 0.073 | 0.196 | 0.017 | 0.360 |
| car | 2915 | 1.300 | 1.885 | 0.195 | 0.000 | 6.279 |
| gif | 2915 | 1.256 | 2.694 | 0.379 | 0.000 | 35.407 |
| eco | 2915 | 10.760 | 0.565 | 10.739 | 9.433 | 12.065 |
| huma | 2915 | 5.741 | 0.890 | 5.864 | 2.890 | 7.272 |
| sci | 2915 | 0.017 | 0.016 | 0.012 | 0.001 | 0.084 |
| out | 2915 | 0.003 | 0.003 | 0.002 | 0.000 | 0.018 |
| ind | 2915 | 0.431 | 0.099 | 0.426 | 0.000 | 0.839 |
| fin | 2915 | 1.067 | 0.646 | 0.886 | 0.118 | 9.623 |
| infra | 2915 | 0.810 | 3.395 | 0.370 | 0.020 | 104.260 |
| gov | 2915 | 0.203 | 0.103 | 0.175 | 0.044 | 0.916 |

a mean value of 0.412 and a standard deviation of 0.150, indicating that there is some fluctuation in the degree of air pollution among different cities. The lowest value is 0.116 and the highest is 0.868. The median of 0.383 suggests that the air pollution level in most cities is below the average level, indicating that the air quality in certain cities may be much lower than in other areas. The core explanatory variable dfi (Digital Financial Inclusion Index) has an average value of 0.185 and a standard deviation of 0.073, showing that the level of development of digital financial inclusion is relatively consistent across the sample cities, although there is a range of variation from 0.017 to 0.360. The median value of 0.196, which is slightly higher than the average value, implies that more than half of the cities have a higher level of digital financial inclusion.

# 4 Results

## 4.1 Baseline regression results

Table 3 shows the test results of the impact of digital inclusive finance on air pollution. Column (1) presents the regression results without control variables, column (2) includes fixed effects, column (3) includes control variables but not fixed effects, and column (4) includes

**Table 3. Baseline regression results.**

|  | (1) | (2) | (3) | (4) |
|---|---|---|---|---|
|  | poll | poll | poll | poll |
| dfi | -0.866*** | -1.620*** | -0.783*** | -1.465*** |
|  | (0.035) | (0.136) | (0.039) | (0.147) |
| eco |  |  | 0.009 | -0.010 |
|  |  |  | (0.007) | (0.006) |
| huma |  |  | 0.094*** | -0.045 |
|  |  |  | (0.003) | (0.028) |
| sci |  |  | -1.495*** | -0.506*** |
|  |  |  | (0.175) | (0.118) |
| out |  |  | 4.693*** | -2.663*** |
|  |  |  | (0.850) | (0.556) |
| ind |  |  | -0.229*** | -0.142*** |
|  |  |  | (0.029) | (0.021) |
| fin |  |  | 0.004 | -0.003 |
|  |  |  | (0.004) | (0.003) |
| infra |  |  | -0.001* | -0.000 |
|  |  |  | (0.001) | (0.000) |
| gov |  |  | -0.123*** | -0.026 |
|  |  |  | (0.032) | (0.029) |
| _cons | 0.573*** | 0.712*** | 0.053 | 1.132*** |
|  | (0.007) | (0.025) | (0.080) | (0.173) |
| Control | No | No | YES | YES |
| City_FE | No | YES | No | YES |
| Year_FE | No | YES | No | YES |
| Obs | 2915 | 2915 | 2915 | 2915 |
| r2 | 0.178 | 0.937 | 0.516 | 0.940 |

*Note:

*, **, and *** indicate statistical significance at the 10%, 5%, and 1% levels, respectively. Standard errors are in parentheses, the same applies below

both control variables and fixed effects. The coefficient of dfi is significantly negative across all specifications, indicating that the development of digital inclusive finance contributes to the reduction of PM2.5 concentration, hence improving air quality. This validates research hypothesis H1. Specifically, digital inclusive finance facilitates the investment in green technologies and sustainable projects by increasing the availability and accessibility of financial services, such as supporting the development and application of clean energy and energy-saving technologies, which directly help to reduce the emission of harmful substances in the air [72]. Additionally, digital inclusive finance may raise public awareness about environmental issues, enhancing the demand for eco-friendly products and services among consumers and businesses and thereby promoting a market transformation towards environmentally friendly products [85].

Considering the unique economic and social context of China and the study region, digital inclusive finance plays a significant role in increasing the penetration rate of financial services and supporting environmental projects. Therefore, policymakers can use digital inclusive finance as a tool to improve environmental quality and encourage research and technological innovation in relevant fields.

## 4.2 Robustness test results

**4.2.1 Replacing the core explanatory variable.** To enhance the robustness of the regression results, a series of robustness tests are conducted. Given the close relationship between the development level of digital finance and the digital economy, this study uses the digital economy to re-measure digital inclusive finance, following the research of Deng and Zhang (2021) [86]. Specifically, the city-level digital economy level (dfi2) is calculated using the entropy method, as per Zhao et al. (2020) [87]. The regression result, shown in column 1 of Table 4, indicates that the coefficient of dfi2 is significantly negative, confirming the robustness of the results.

**4.2.2 Excluding municipality samples.** Considering the unique political status and urban administrative structure of municipalities, which might specifically influence the study results, the study excludes municipality samples and re-conducts the regression. The results, shown in

**Table 4. Robustness test results.**

|  | (1) | (2) | (3) | (4) |
|---|---|---|---|---|
|  | **poll** | **poll** | **poll** | **poll** |
| dfi |  | -1.445*** |  |  |
|  |  | (0.147) |  |  |
| dfi2 | -0.854*** |  |  |  |
|  | (0.296) |  |  |  |
| Ldfi |  |  | -0.001*** |  |
|  |  |  | (0.000) |  |
| did |  |  |  | -0.019*** |
|  |  |  |  | (0.003) |
| _cons | 1.067*** | 1.178*** | 1.013*** | 0.945*** |
|  | (0.176) | (0.171) | (0.184) | (0.177) |
| Control | YES | YES | YES | YES |
| City_FE | YES | YES | YES | YES |
| Year_FE | YES | YES | YES | YES |
| Obs | 2915 | 2871 | 2649 | 2915 |
| r2 | 0.938 | 0.941 | 0.937 | 0.938 |

column (2) of Table 4, with a significantly negative dfi coefficient, confirm the robustness of the conclusions.

**4.2.3 Lagged treatment.**    Considering the potential time lag in the impact of digital inclusive finance on air pollution and to reduce the interference of causality, the core explanatory variable (Ldfi) and control variables are lagged by one period. The results, as shown in column (3) of Table 4, indicate that the coefficient of lagged digital inclusive finance remains significantly negative, confirming the robustness of the conclusions.

**4.2.4 Exogenous shock test.**    To robustly assess whether digital inclusive finance promotes air quality improvement, this paper uses the pilot of the "National Big Data Comprehensive Experimental Zone" as an exogenous policy shock and employs the Difference-in-Differences (DID) method to assess this practical issue. The major task of the Big Data Experimental Zone is to achieve data sharing and utilization, providing significant support for the development of digital inclusive finance. The expansionary nature of the pilot policy provides a good quasi-natural experiment research strategy. Specifically, cities included in the first and second batches of the national-level Big Data Comprehensive Experimental Zone in 2015 and 2016 are selected as the treatment group, and the remaining cities as the control group, constructing the following multi-period DID model.

$$poll_{i,t} = \alpha_0 + \alpha_1 did_{i,t} + \delta X + \gamma_i + \omega_t + \varepsilon_{i,t} \tag{6}$$

The did coefficient is significantly negative, as shown in column (4) of Table 4, indicating that digital inclusive finance still promotes air quality improvement. The validity of the DID model depends on the absence of time trend differences between the treatment and control groups before the policy shock. The parallel trend test results, as shown in Fig 1, indicate that the coefficient estimates before policy implementation are mostly insignificant, while they

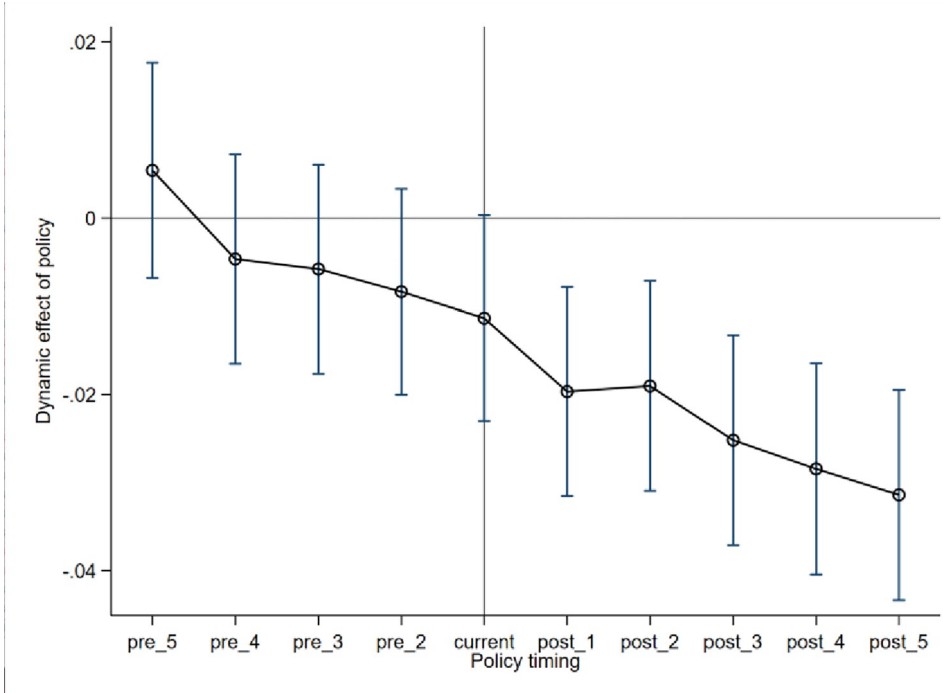

**Fig 1. Parallel trend test results.**

show a significant downward trend after policy implementation, proving the parallel trend assumption.

**4.2.5 Instrumental variable test.**   Furthermore, this paper uses the instrumental variable method to address potential reverse causality issues in the model. Following Bartik (2009) [88], a "Bartik instrument" is constructed, i.e., the product of the lagged digital inclusive finance and the first difference of digital inclusive finance as an instrumental variable (iv). The Bartik instrumental variable is directly related to the local level of digital inclusive finance development but does not directly affect urban environmental pollution through other channels. This means that the Bartik instrumental variable can address potential endogeneity issues between digital inclusive finance development and urban environmental pollution. By introducing the Bartik instrumental variable and using the two-stage least squares method (2SLS) for estimation, the results are shown in Table 5. Column (1) shows the first-stage regression results, where the iv coefficient is significantly positive, indicating relevance between digital inclusive finance and the instrumental variable. Additionally, the F-statistic value of 26.63 suggests that the weak instrument variable test is passed. Column (2) shows the second-stage regression results, where the dfi coefficient remains significantly negative, further confirming the robustness of the conclusions.

## 4.3 Mechanism test results

**4.3.1 Public environmental awareness mechanism test.**   To test whether digital inclusive finance can improve air quality by enhancing public environmental awareness, regression is conducted on mechanism test models (2) and (3). The regression results, as shown in columns (1) and (2) of Table 6, indicate that in column (1), the coefficient of dfi is significantly positive, suggesting that digital inclusive finance significantly promotes the enhancement of public environmental awareness. This might be due to the widespread accessibility of digital finance, which makes environmental information more readily available to the public, enhancing environmental consciousness and encouraging participation in eco-friendly behaviors. In column (2), the coefficient of car is significantly negative, implying that public environmental awareness significantly improves air quality. This further indicates that public concern for environmental issues contributes to more effective environmental protection measures, thus improving air quality. The mechanism test results confirm that digital inclusive finance indeed improves air quality by enhancing public environmental awareness, validating research hypothesis H2.

This finding aligns with existing environmental economics theories, which state that public participation and awareness are crucial for the success of environmental protection. In the

**Table 5. Instrumental variable regression results.**

|  | (1) | (2) |
|---|---|---|
|  | **dfi** | **poll** |
| dfi |  | -7.208*** |
|  |  | (1.607) |
| iv | 2.928*** |  |
|  | (0.490) |  |
| Control | YES | YES |
| City_FE | YES | YES |
| Year_FE | YES | YES |
| Obs | 2650 | 2650 |

**Table 6. Mechanism test results.**

|  | (1) | (2) | (3) | (4) |
|---|---|---|---|---|
|  | car | poll | gif | poll |
| dfi | 19.164*** |  | 63.455*** |  |
|  | (2.886) |  | (3.978) |  |
| car |  | -0.011*** |  |  |
|  |  | (0.001) |  |  |
| gif |  |  |  | -0.002*** |
|  |  |  |  | (0.001) |
| _cons | -2.660 | 1.058*** | 7.251 | 1.103*** |
|  | (3.407) | (0.172) | (4.696) | (0.176) |
| Control | YES | YES | YES | YES |
| City_FE | YES | YES | YES | YES |
| Year_FE | YES | YES | YES | YES |
| Obs | 2915 | 2915 | 2915 | 2915 |
| r2 | 0.852 | 0.940 | 0.862 | 0.938 |

digital era, the development of financial technology provides new channels and opportunities to raise public environmental awareness, thereby playing a positive role in environmental protection. These results not only reinforce the understanding of the impact mechanism of digital inclusive finance but also provide important insights for formulating effective environmental policies, especially in promoting public participation and raising environmental awareness.

**4.3.2 Green technology innovation mechanism test.** To test whether digital inclusive finance can improve air quality by enhancing the level of green technology innovation, regression is conducted on mechanism test models (2) and (3). The results, as shown in columns (3) and (4) of Table 6, indicate that in column (3), the coefficient of dfi is significantly positive, suggesting that digital inclusive finance significantly promotes the level of green technology innovation. A possible explanation is that digital inclusive finance, by providing more convenient and diverse financing channels, lowers the funding threshold for green technology innovation, promoting R&D activities and technological innovation. In column (4), the coefficient of gif is significantly negative, indicating that green technology innovation significantly improves air quality. This suggests that green technology innovation plays a key role in increasing energy efficiency and reducing pollution emissions. These innovations might include clean energy solutions, efficient energy management systems, and pollution reduction technologies, all contributing to improved air quality. The mechanism test results confirm that digital inclusive finance also improves air quality by enhancing green technology innovation, validating research hypothesis H3.

This finding is also consistent with modern environmental protection theories, where technological innovation is considered a key factor in solving environmental issues and enhancing sustainability. Digital inclusive finance, as an important tool for promoting technological innovation, accelerates the development and application of green technologies by providing funding and facilitating information exchange. The implementation of these technologies is crucial for achieving more environmentally friendly and sustainable development.

## 4.4 Spatial spillover effect results

To verify the spatial spillover effect of digital inclusive finance on air pollution, regression is conducted on the spatial spillover effect test model. Prior to regression, it is generally necessary to perform a spatial autocorrelation test on the dependent variable and the core explanatory

**Table 7. Spatial autocorrelation test results.**

|  | poll | | | dfi | | |
|---|---|---|---|---|---|---|
| year | Moran's I | Z value | P value | Moran's I | Z value | P value |
| 2011 | 0.2244 | 41.6952 | 0.0000 | 0.1129 | 21.3134 | 0.0000 |
| 2012 | 0.2244 | 41.7311 | 0.0000 | 0.1239 | 23.3437 | 0.0000 |
| 2013 | 0.2361 | 43.8263 | 0.0000 | 0.1206 | 22.7347 | 0.0000 |
| 2014 | 0.2113 | 39.3210 | 0.0000 | 0.1001 | 18.9940 | 0.0000 |
| 2015 | 0.2487 | 46.1694 | 0.0000 | 0.1087 | 20.5617 | 0.0000 |
| 2016 | 0.2534 | 47.0513 | 0.0000 | 0.1056 | 19.9827 | 0.0000 |
| 2017 | 0.2342 | 43.5070 | 0.0000 | 0.1238 | 23.3229 | 0.0000 |
| 2018 | 0.2472 | 45.8781 | 0.0000 | 0.1510 | 28.3049 | 0.0000 |
| 2019 | 0.2502 | 46.4227 | 0.0000 | 0.1550 | 29.0361 | 0.0000 |
| 2020 | 0.2638 | 48.8896 | 0.0000 | 0.1645 | 30.7564 | 0.0000 |
| 2021 | 0.2417 | 44.8449 | 0.0000 | 0.1738 | 32.4718 | 0.0000 |

variable. The global Moran's I test for air pollution and digital inclusive finance, as shown in Table 7, indicates significant positive values for both, suggesting spatial autocorrelation and clustering phenomena. This means air pollution and digital inclusive finance show a non-random distribution pattern in geographic space, reflecting the importance of geographic location in understanding their relationship. This spatial clustering may be related to similar economic development levels, environmental policies, or industrial structures, leading to similar spatial distribution characteristics in neighboring areas.

An appropriate spatial econometric model can more accurately reflect the reasons for spatial dependence among geographical economic entities and further explore the impact of different spatial association mechanisms. Following the method outlined by Elhorst (2014) [89] and Han & Xie (2017) [90], we performed a spatial econometric analysis following the "specific-to-general" approach that initially considers the sample as cross-sectional data and employs ordinary least squares (OLS) for preliminary regression. To determine the appropriateness of the Spatial Autoregressive (SAR) or Spatial Error Model (SEM), we utilized the Lagrange Multiplier (LM) tests. The SAR model was chosen if the LM-lag test was significant; alternatively, the SEM model was used if the LM-error was significant. In instances where both tests were significant, we further compared R-LM-lag and R-LM-error. Based on this comparison, if R-LM-lag was significant but R-LM-error was not, we adopted the SAR model. Conversely, if the results favored the SEM model, then it was selected. When both tests indicate significance, the Spatial Durbin Model (SDM) was considered. If the non-spatiality of the model was rejected, the "general-to-specific" approach was followed to further test for fixed effects in the SDM model using a likelihood ratio method. We applied the Hausman test to determine whether to estimate using fixed or random effects. Additionally, we used Wald or LM tests to assess whether the SDM model reduces to the SAR or SEM model. If the SDM model was not simplified to SAR or SEM, as indicated by both tests being rejected, it was retained because of its ability to incorporate both spatial autoregressive and spatial error components. The optimal selection indicated by the results in Table 8 was the SDM model with dual time-space fixed effects, making it the most suitable for our spatial econometric analysis.

The further results from the Spatial Durbin Model (SDM) regression are displayed in column (1) of Table 9, where we can see that the coefficient for dfi (Digital Financial Inclusion Index) remains significantly negative. Additionally, both the spatial Durbin term and the indirect effect term's coefficients are significantly negative, indicating that the development of digital inclusive finance can significantly reduce air pollution levels in neighboring areas, thus

**Table 8. Spatial model test results.**

| Test | Value | Test | Value |
|---|---|---|---|
| LM-lag | 5074.315*** | Hausman test | 93.32*** |
| R-LM-lag | 80.563*** | Wald_spatial_lag | 179.60*** |
| LM-error | 2.9e+04*** | Wald_spatial_error | 228.92*** |
| R-LM-error | 2.4e+04*** | LR_spatial_lag | 174.67*** |
| LR-test-id | 82.60*** | LR_spatial_error | -912.90*** |
| LR-test-time | 4695.50*** | | |

confirming research hypothesis H4. For robustness, this paper also includes the regression results for the SAR (Spatial Autoregressive) and SEM (Spatial Error Model), as shown in columns (2) and (3) of Table 9, and the results show no significant difference. Specifically, digital inclusive finance may reduce air pollution in neighboring areas through the promotion of environmental technologies and practices within the region, enhancing regional environmental awareness, and altering behavioral patterns, leading to positive effects in adjacent areas. This suggests an important geographical spillover effect, where the development of digital inclusive finance in one area positively impacts the environment not only locally but also promotes improvements in air quality in neighboring regions through spatial linkages.

This discovery provides a new perspective for understanding the environmental effects of digital inclusive finance and offers important references for regional environmental policy formulation and future research. Particularly in considering regional economic interdependence and coordinated environmental policies, these findings indicate that regional environmental protection efforts should be viewed as interconnected and mutually influencing systems.

## 4.5 Non-linear test results

To test whether the impact of digital inclusive finance on air pollution exhibits non-linear characteristics, regression is conducted on the non-linear effect test model. Specifically, digital inclusive finance is used as a threshold variable for threshold testing. The results, as shown in Table 10, reveal that both single and double thresholds pass the test, while the triple threshold

**Table 9. Spatial Durbin model regression results.**

| | (1) | (2) | (3) |
|---|---|---|---|
| | Poll | Poll | Poll |
| dfi | -0.841*** | -1.161*** | -1.066*** |
| | (0.137) | (0.116) | (0.116) |
| W*dfi | -2.623*** | | |
| | (0.814) | | |
| Dfi (Direct) | -1.414*** | -1.432*** | |
| | (0.251) | (0.377) | |
| Dfi (Indirect) | -150.228** | -71.199 | |
| | (60.992) | (93.813) | |
| Dfi (Total) | -151.642** | -72.631 | |
| | (61.204) | (94.172) | |
| Control | YES | YES | YES |
| City_FE | YES | YES | YES |
| Year_FE | YES | YES | YES |
| Obs | 2915 | 2915 | 2915 |
| r2 | 0.200 | 0.198 | 0.214 |

**Table 10. Threshold test results.**

| Threshold variable | Threshold test | F value | P value | Critical value | | |
|---|---|---|---|---|---|---|
| | | | | 1% | 5% | 10% |
| dfi | Single threshold test | 17.32** | 0.0220 | 21.2094 | 14.6140 | 12.5703 |
| | Double threshold test | 18.07** | 0.0280 | 20.7046 | 15.0482 | 12.8989 |
| | Triple threshold test | 16.08 | 0.4660 | 42.3968 | 33.3015 | 27.8679 |

does not, suggesting that a double threshold model is a more appropriate analytical tool. Specifically, the first threshold value is set at 0.0369, and the second at 0.1410.

The panel double threshold model regression results, as shown in Table 11, show that when the value of digital inclusive finance is below 0.0369, its impact on air pollution (dfi coefficient) is -2.487, indicating a strong negative effect. However, when the value of digital inclusive finance is between 0.0369 and 0.1410, this coefficient drops to -1.578. Further, when the value of digital inclusive finance exceeds 0.1410, the dfi coefficient drops further to -1.452. These results suggest that as digital inclusive finance develops, its impact on reducing air pollution gradually weakens, validating research hypothesis H5.

This discovery of non-linear effects aligns with existing economic theories. In economics, diminishing marginal utility is a core concept, meaning that the additional utility (or benefit) decreases as consumption or production increases. In the relationship between digital inclusive finance and air pollution, this indicates that initial financial development has a larger impact on the environment, but as development progresses, the additional environmental benefits gradually diminish. This could be due to saturation in environmental technology innovation and application in the later stages of digital inclusive finance development, or the energy consumption associated with digitalization begins to exert pressure on the environment. Overall, the revelation of this non-linear effect provides a new perspective for understanding the role of digital inclusive finance in environmental protection and offers important guidance for formulating effective environmental and financial policies.

## 4.6 Heterogeneity analysis results

**4.6.1 Impact of environmental regulation intensity.** In exploring the relationship between digital inclusive finance and air pollution, the varying intensity of environmental

**Table 11. Panel threshold model regression results.**

| | (1) |
|---|---|
| | poll |
| dfi (dfi≤0.0369) | -2.487*** |
| | (0.269) |
| dfi (0.0369<dfi≤0.1410) | -1.578*** |
| | (0.148) |
| dfi (dfi>0.1410) | -1.452*** |
| | (0.144) |
| _cons | 1.149*** |
| | (0.167) |
| Control | YES |
| City_FE | YES |
| Year_FE | YES |
| Obs | 2915 |
| r2 | 0.795 |

**Table 12. Heterogeneity analysis results.**

|  | (1) | (2) | (3) | (4) | (5) |
|---|---|---|---|---|---|
|  | poll | poll | poll | poll | poll |
| dfi | -1.641*** | -0.799*** | -0.666*** | -1.614*** | -0.727*** |
|  | (0.209) | (0.226) | (0.252) | (0.252) | (0.211) |
| _cons | 1.186*** | 1.135*** | 1.042*** | 1.224*** | 1.356*** |
|  | (0.226) | (0.284) | (0.334) | (0.275) | (0.187) |
| Control | YES | YES | YES | YES | YES |
| City_FE | YES | YES | YES | YES | YES |
| Year_FE | YES | YES | YES | YES | YES |
| Obs | 1552 | 1334 | 1100 | 1100 | 715 |
| r2 | 0.946 | 0.947 | 0.951 | 0.943 | 0.902 |

policy enforcement across different regions may significantly influence the results. Environmental regulation, as a key factor affecting corporate and individual behaviors, plays a crucial role in environmental protection and pollution control. This study, following the method of Chen and Chen (2018) [91], uses Python to process city government work reports, tallying the frequency of keywords related to environmental regulation (such as environmental protection, eco, pollution, energy consumption, emission reduction, sewage, ecology, green, low-carbon, air, chemical oxygen demand, sulfur dioxide, carbon dioxide, PM10, and PM2.5). Cities are classified into high and low environmental regulation intensity groups based on the proportion of environmental regulation keywords to total word frequency. If a city's environmental regulation intensity is equal to or greater than the annual median of all cities in the province, it is considered a high environmental regulation city; otherwise, it is considered a low environmental regulation city. The regression results for high and low environmental regulation intensity samples, as shown in columns (1) and (2) of Table 12, indicate that in cities with higher environmental regulation intensity, the positive impact of digital inclusive finance on air pollution is more significant. This may be due to stronger environmental regulations providing a more favorable policy environment for the environmental benefits brought by digital inclusive finance, enhancing its support effect on eco-friendly projects.

This analysis provides important insights for policymakers. When formulating environmental policies, the importance of environmental regulation intensity in strengthening the role of digital inclusive finance in environmental protection should be considered. Strengthening environmental regulation can not only directly improve air quality but also enhance the environmental effect of digital inclusive finance, achieving more comprehensive and sustainable environmental improvement.

### 4.6.2 Impact of geographic location

Geographic location's distinct characteristics may significantly impact the study of the relationship between digital inclusive finance and air pollution. China's eastern, central, and western regions differ markedly in economic development levels, industrial structures, environmental policy enforcement, and natural environments, potentially affecting the impact of digital inclusive finance on air pollution. To explore the influence of geographic location, the study first divides the total sample into eastern, central, and western city samples based on China's geographical distribution and conducts subgroup regressions. This grouping reflects the economic and environmental characteristics of different regions, providing a basis for further heterogeneity analysis. The regression results for eastern, central, and western samples, as shown in columns (3), (4), and (5) of Table 12, indicate that the impact of digital inclusive

**Table 13. Extended analysis results.**

| | (1) | (2) |
|---|---|---|
| | poll | poll |
| dfi | -1.406*** | -1.399*** |
| | (0.147) | (0.152) |
| did1 | 0.020*** | |
| | (0.007) | |
| dfidid1 | -0.120*** | |
| | (0.030) | |
| did2 | | -0.001 |
| | | (0.015) |
| dfidid2 | | -0.044 |
| | | (0.060) |
| _cons | 1.213*** | 1.128*** |
| | (0.174) | (0.173) |
| Control | YES | YES |
| City_FE | YES | YES |
| Year_FE | YES | YES |
| Obs | 2915 | 2915 |
| r2 | 0.940 | 0.940 |

finance on air pollution improvement is most significant in the central region, followed by the western region, with the smallest effect in the eastern region. This may reflect differences in economic development levels and environmental policy enforcement across regions. For example, central and western regions, due to their lower economic development levels, may rely more on digital inclusive finance as a tool for environmental protection. In contrast, the eastern region, with faster economic development, has a more diversified array of pollution control measures, resulting in a relatively lower marginal utility of digital inclusive finance. Additionally, this finding is consistent with the study's non-linear effect test results, where initial financial development in less economically developed areas may bring greater environmental improvement benefits, but these additional environmental benefits gradually decrease with deeper development. This effect may be more pronounced in economically developed areas, as they might have already implemented various environmental control measures, leading to a lower marginal utility for digital inclusive finance.

## 4.7 Extended analysis

In discussing the impact of digital inclusive finance on environmental protection, understanding the relevant policy context is crucial for deepening the conclusions of this study. This research explores how related policies affect the relationship between the explanatory variable (digital inclusive finance) and the dependent variable (air pollution). Particularly in a rapidly changing policy environment, studying how these policy backgrounds shape and motivate the relationship between digital inclusive finance and environmental protection helps to more comprehensively understand this complex dynamic.

**4.7.1 Testing the incentive effect of the "Smart City" pilot policy.** The Smart City pilot policy, a significant component of China's national plans, aims to modernize cities through informatization and digitalization. In the context of the current technological revolution driven by big data, artificial intelligence, and cloud computing, this policy intends to use digital technologies to optimize urban operations, improve residents' quality of life, and maintain

economic competitiveness. As Wei and Ma (2022) [92] observed, the construction of smart cities has become an irreversible trend in the wave of technology development. This section preliminarily explores whether this policy influences the effect of digital inclusive finance on air pollution.

Considering China's Smart City pilot construction was carried out in three batches in 2012, 2013, and 2014, cities included in these three batches are used as the experimental group, while the remaining cities serve as the control group. A new multi-period Difference-in-Differences (DID) model is constructed to test whether the Smart City pilot policy affects the impact of digital inclusive finance on air pollution. As shown in the first column of Table 13, the interaction term between policy dummy variable (did1) and digital financial inclusion (dfidid1) is significantly negative, and has a positive impact on air quality, indicating a significant positive incentive effect of the Smart City pilot policy in the mechanism of digital inclusive finance affecting air pollution.

This result is instructive for urban planners and policymakers. It suggests that the construction of smart cities can be an effective policy tool, not only improving urban efficiency and residents' quality of life but also helping to promote environmental protection through digital inclusive finance. Therefore, policymakers should pay attention to the interplay between smart city construction and digital inclusive finance, coordinating policies in these two areas to achieve more sustainable urban development.

**4.7.2 Testing the incentive effect of the "major protection, no major development" requirement.** The Yangtze River Economic Belt is one of China's national regional strategies. The Yangtze River, the lifeline of the Chinese nation, faced severe ecological challenges due to long-term excessive and disorderly development. In early 2016, President Xi Jinping proposed the strict requirement of "major protection, no major development" at a symposium in Chongqing to promote the development of the Yangtze River Economic Belt. In the following years, cities in the Yangtze River Economic Belt experienced lower but higher-quality economic growth, significantly improving the ecological environment of the Yangtze River basin. This section explores whether the "major protection, no major development" requirement affects the impact of digital inclusive finance on air pollution, contributing to understanding the influence of digital inclusive finance on environmental protection under different policy environments.

The study uses cities in the Yangtze River Economic Belt as the experimental group and other cities as the control group, constructing a new multi-period DID model to test the effect of the "major protection, no major development" requirement. As shown in the second column of Table 13, the interaction term between the policy dummy variable (did2) and digital financial inclusion (dfidid2) is negative but not significant, and has a positive impact on air quality, indicating a certain positive incentive effect of the "major protection, no major development" requirement.

This result offers important insights for policymakers, emphasizing the necessity of ecological protection while promoting regional economic development, especially in ecologically fragile areas. Digital inclusive finance can play a role in this process, particularly in supporting green technology and projects and promoting environmental protection.

The study reveals that at the explanatory variable level, policies such as the Smart City pilot show more significant influence compared to policies at the dependent variable level, like the "major protection, no major development" requirement. A possible explanation is that policies at the dependent variable level, such as environmental regulation in the Yangtze River Economic Belt, positively impact the environment but not solely through the pathway of digital inclusive finance. These policies, through comprehensive environmental governance measures like pollution emission control and ecological protection zone construction, improve regional

environmental quality. In contrast, environmental policies at the dependent variable level utilize diversified environmental improvement methods, not limited to financial and technological innovation, but also including policy formulation, public awareness, and industry standards. Additionally, policies at the dependent variable level usually have a more comprehensive and long-term vision, focusing on sustainable development and laying the foundation for future environmental protection through long-term planning and sustained efforts. This finding indicates that while policies at the dependent variable level play a key role in environmental protection, their impact is not solely achieved through the pathway of digital inclusive finance. Instead, these policies collectively contribute to environmental improvement through multiple channels and measures, showcasing the complexity of integrating environmental policies across various domains. Policies at the explanatory variable level, particularly the Smart City pilot policy, provide more direct and focused support for the application of digital inclusive finance in environmental protection by directly influencing the fields of digital finance and technology. Therefore, the results of this study emphasize the need for comprehensive consideration of the interaction and combined impact of different policy levels when formulating environmental policies. This understanding provides an important perspective and guidance for developing more comprehensive and effective environmental protection strategies.

## 5 Discussion

This paper delves into the impact of digital inclusive finance on air pollution, yielding many significant results. First, the findings reveal the crucial role of digital inclusive finance in significantly mitigating air pollution [10, 56, 93]. This discovery highlights the potential of digital inclusive finance in supporting environmental projects and fostering the development of green technologies through the provision of convenient financial services. It suggests that digital inclusive finance is not merely a tool to enhance financial accessibility but can also be a significant force in promoting environmental sustainability [17, 26]. These results offer a new perspective on the interaction between financial technology and environmental protection, emphasizing the crucial role of financial innovation in environmental conservation. By directing funds toward eco-friendly projects and green technologies, digital inclusive finance contributes to alleviating air pollution, thereby improving environmental quality [5, 59].

In discussing the mechanisms through which digital inclusive finance improves air quality, the paper identifies two key channels: enhancing public environmental awareness and fostering green technology innovation. Digital inclusive finance, as an innovative form of financial service, not only provides financial support but also, through its wide platform and network, effectively raises public awareness and participation in environmental issues [56, 94]. Simultaneously, it encourages the research and application of green technologies by simplifying financial processes and reducing financing costs [26, 59]. These findings underscore the role of financial services in transforming societal ideologies and technological innovation, demonstrating that financial tools can effectively serve as drivers for environmental protection.

The exploration of the spatial spillover effects of digital inclusive finance reveals that this effect is not merely a geographical expansion of financial services but a broader socio-economic phenomenon. This spillover might result from digital inclusive finance promoting the spread of environmental technologies and practices within a region, thereby influencing the environmental consciousness and behavioral patterns across the area [46, 95]. For example, successful environmental projects in one region might gain recognition and emulation in neighboring areas through digital platforms, creating a ripple effect. On the other hand, the discovery of non-linear effects unveils the complexity of the relationship between digital inclusive finance and environmental protection. Initially, digital inclusive finance might rapidly

promote the adoption of eco-friendly technologies and behavior changes, but over time, this effect may diminish as initial innovations become widely adopted, and new environmental challenges require novel solutions [5, 32]. This suggests a continuous push for financial innovation and technological advancement is necessary for sustained environmental quality improvement.

The heterogeneity analysis, considering both geographic location and environmental regulation intensity, further enriches the understanding of the relationship between digital inclusive finance and environmental protection. The significance of the central region emphasizes the urgency of environmental protection during economic transition, while the varying performances of the eastern and western regions reveal the roles of economic development stages and regional characteristics in environmental conservation [18, 44]. Additionally, the analysis of environmental regulation intensity indicates that in cities with stricter environmental regulations, digital inclusive finance is more effective in improving air quality, likely due to a more proactive response to environmental policies in these areas [59, 96]. This regional variation provides robust data support for formulating more nuanced and region-specific environmental and financial policies, emphasizing the importance of considering regional economic development levels and environmental regulation intensity in environmental policy formulation.

In the analysis of policy incentive effects, the study finds that both the Smart City pilot policy and the "major protection, no major development" requirement significantly impact the relationship between digital inclusive finance and environmental protection. Especially the Smart City pilot policy, by promoting urban digitalization and informatization, enhances the application effect of digital inclusive finance in the environmental domain. This indicates that policy environments and technological advancements can jointly influence the environmental benefits of digital inclusive finance, strengthening its role in promoting sustainable development. The "major protection, no major development" requirement, while enhancing the environmental impact of digital inclusive finance, has a relatively limited effect, possibly due to the policy's greater emphasis on ecological protection rather than financial innovation [97, 98]. These findings provide empirical support for policymakers considering the synergistic effects of environmental protection and financial technology development.

This paper makes important contributions and innovations to the theoretical framework for understanding the interaction between digital inclusive finance and environmental protection. Firstly, the paper offers a new perspective on the interplay between financial technology and environmental protection, with existing literature primarily focusing on the impact of digital inclusive finance on economic development and financial inclusion, while its environmental effects have been less studied. By deeply analyzing the relationship between digital inclusive finance and air pollution, this study not only considers the direct impact on air pollution but also the indirect effects through heightened public environmental awareness and green technology innovation. Secondly, the paper reveals the spatial spillover effects and non-linear characteristics of digital inclusive finance on the improvement of air pollution. It shows that its positive impacts can transcend geographical boundaries and present different environmental effects at different stages of development. Thirdly, heterogeneity analysis delves into the influence of geographical location and environmental regulatory intensity on the environmental effects of digital inclusive finance. It uncovers the role of economic development stages and regional characteristics in environmental protection, which is crucial for formulating more fine-tuned and regionally adapted environmental and financial policies. Fourthly, the analysis of policy incentive effects finds that smart city pilot policies and the policy imperative "to coordinate protection and to avoid excessive development" have a significant impact on the relationship between digital inclusive finance and environmental protection. This provides empirical support for policymakers considering the synergistic effects of environmental

protection and financial technology development. In conclusion, this paper not only enriches the theoretical research on digital inclusive finance and environmental protection but also provides concrete guidance to policymakers and practitioners on how to utilize digital financial tools to promote environmental protection, thus bridging the gap between theory and practice.

In discussing the limitations and future directions of the study, it's important to recognize the potential limitations of sample selection and model settings. As the study uses city-level data, it may not fully capture the detailed changes at the micro level. Additionally, although the model settings consider spatial effects and non-linear characteristics, there may still be other unobserved confounding factors. Future research should expand the sample scope, covering a broader range of regions and diverse types of environmental pollution. Exploring the interactions of digital inclusive finance with other socio-economic factors and more detailed mechanism analyses will provide a more comprehensive perspective on its impact on the environment. Moreover, considering the rapid development of digital inclusive finance, future research could focus on its long-term impacts and the differences in its effects at different stages of economic development.

## 6 Conclusion

In the context of the current global economic and technological environment, the development of digital inclusive finance has not only reshaped the financial services sector but also profoundly impacted environmental protection. This paper conducts a comprehensive analysis of the relationship between digital inclusive finance and air pollution. The results indicate that digital inclusive finance has a significant positive impact on reducing air pollution. This is primarily due to digital inclusive finance fostering investment in green technologies and sustainable projects and raising public awareness of environmental issues, thereby effectively improving air quality. Additionally, the study finds a spatial spillover effect of digital inclusive finance on air pollution improvement, suggesting that its positive impacts can transcend geographical boundaries and influence neighboring areas. The research also reveals the non-linear characteristics of digital inclusive finance's impact on air pollution, showing a "diminishing marginal utility" trend in its contribution to environmental protection as it develops. Furthermore, heterogeneity analysis results demonstrate that the environmental effects of digital inclusive finance are significantly influenced by different regional economic and environmental backgrounds. Lastly, the study of the Smart City pilot policy and the "major protection, no major development" requirement emphasizes the importance of policies in strengthening the link between digital inclusive finance and environmental protection. Particularly, the Smart City policy plays a more significant positive role in strengthening this connection. Overall, this study provides a new perspective on the role of digital inclusive finance in environmental protection and offers important guidance for formulating effective environmental and financial policies.

## Author Contributions

**Conceptualization:** Zexing Wang, Min Fan, Yaojun Fan.

**Data curation:** Min Fan.

**Formal analysis:** Zexing Wang.

**Investigation:** Yaojun Fan.

**Methodology:** Zexing Wang, Min Fan.

**Project administration:** Zexing Wang.

**Resources:** Zexing Wang, Yaojun Fan.

**Software:** Min Fan.

**Supervision:** Yaojun Fan.

**Visualization:** Min Fan.

**Writing – original draft:** Zexing Wang, Min Fan, Yaojun Fan.

**Writing – review & editing:** Zexing Wang, Min Fan.

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
