## [Decision Letter · Decision Letter 0]

22 Apr 2024

PONE-D-24-05850The Impact of Digital Inclusive Finance on Environmental Pollution: A Case Study of Air PollutionPLOS ONE

Dear Dr. Fan,

Thank you for submitting your manuscript to PLOS ONE. After careful consideration, we feel that it has merit but does not fully meet PLOS ONE’s publication criteria as it currently stands. Therefore, we invite you to submit a revised version of the manuscript that addresses the points raised during the review process.

We look forward to receiving your revised manuscript.

Kind regards,

Pengyu Chen

Academic Editor

PLOS ONE

Journal Requirements:

3. Please include a copy of Table 13 which you refer to in your text on page 27.

Reviewers' comments:

Reviewer's Responses to Questions

**Comments to the Author**

1. Is the manuscript technically sound, and do the data support the conclusions?

Reviewer #1: Yes

Reviewer #2: Yes

2. Has the statistical analysis been performed appropriately and rigorously? 

Reviewer #1: Yes

Reviewer #2: No

3. Have the authors made all data underlying the findings in their manuscript fully available?

Reviewer #1: Yes

Reviewer #2: No

4. Is the manuscript presented in an intelligible fashion and written in standard English?

Reviewer #1: Yes

Reviewer #2: No

5. Review Comments to the Author

Reviewer #1: This is a recognizable study in which the authors specifically discuss the impact of digital inclusion finance on environmental pollution, reflecting the scientific nature of sufficient workload in terms of data and methodology, and the conclusions are of theoretical significance and practical value. I have the following suggestions or questions about this manuscript for the authors':

1. the authors' approach to the data on PM2.5 and digital inclusion finance seems to be inappropriate. The authors claim to have divided the data for the two indicators by 100 and 1,000 for the sake of data homogeneity and elimination of scale effects (found in lines 374 and 381). However, dividing by 1000 does not eliminate the scale effect, which should be eliminated by dividing by certain economic indicators that reflect the size of the city, such as GDP and population. In order to achieve homogeneity of the data, the natural logarithm should be taken instead of dividing by 100, so I suggest that the authors reconsider the treatment of the data.

2. The discussion of Hypothesis 1 is slightly lengthy and the authors may consider streamlining it to highlight the direct link between digital inclusion finance and environmental pollution.

3. More results should be reported in Table 8, e.g. the authors should test whether SDM is the most appropriate model and consider whether to report regression results from spatial autoregressive models or spatial error models.

Reviewer #2: 1. Insufficient motivation for the study.

2. The digital financial inclusion policy is directed at micro and small enterprises and has little strength of effect on the population. It is therefore recommended to look for more rational mechanism variables.

3. lack of research innovation part.

4. 2.2 part of the logic is confusing and lacks a theoretical framework.

5. It is suggested to update the references.

6. It is suggested to add part 2.3 which has literature on the effect of increasing public awareness of environmental protection and environmental pollution. 7.

7. It is suggested to add control variables related to environmental pollution, such as the level of regional green financial development, green financing capacity, etc. It is also suggested to add the control variables related to environmental pollution. Try to cover economic, social, political and climatic environment to facilitate more accurate identification and alleviate the endogeneity problem of omitted variables.

8. Equations 2-4 lack explanations of relevant variables. and Equation 2 is exactly the same as Equation 1. Please double check equations 3 and 4. could you please provide the DOI number of the paper you are drawing from in your response. Because I didn't find it in the references.

9. This paper contains too many empirical models, it is suggested to selectively eliminate some of them. Because too many models lead to insufficient research analysis.

10. Table 2 lacks analysis.

11. The benchmark regression results in this paper lack the support of existing literature.

12. Please label what the brackets in the regression results represent.

13. Professional English language proofreading is recommended before resubmission.

14. Please check for spelling errors.

Overall, the manuscript is in line with the scope of the study in PLOS ONE, and it is recommended that the manuscript be published after making revisions. We wish you a speedy publication of your paper.

6. PLOS authors have the option to publish the peer review history of their article (what does this mean?). If published, this will include your full peer review and any attached files.

Reviewer #1: No

Reviewer #2: No

---

## [Author Response · Author response to Decision Letter 0]

24 May 2024

Response to Reviewer 1:

Q1: Thank you very much for your concern about our research methods and the suggestions you have provided. We apologize for the incorrect description in the paper regarding the elimination of data heterogeneity and scale effects. This was actually an erroneous statement, and not a measure we have taken in our research. We are grateful that you have pointed this out. In our research, both variables are in level form, which allows for direct comparisons of the data. This comparison is meaningful because it allows us to directly evaluate the interactions and influences between different variables. Changing the data processing method may affect the effectiveness of this direct comparison, thereby affecting the interpretability of the research results. To make the variable selection more credible, we have supplemented more reasons and literature support in the paper to justify our choice. We hope to gain your recognition.

Q2: Thank you for reviewing our paper and providing valuable comments. Regarding the issue you mentioned about the need for further simplification and clarification in Section 2.2, we have comprehensively revised the logic and supplemented the theoretical framework in combination with the suggestions of Reviewer 2. We have carefully considered which content is core and necessary, while simplifying the overly complex discourse to ensure that the entire section is not only logically clear, but also more concise and focused. These improvements aim to improve the overall quality and readability of the paper, making it more in line with the publication standards of the journal.

Q3: Thank you for reviewing our research methods and providing suggestions. Regarding your comments, I would like to confirm that we have carried out a comprehensive and rigorous spatial econometric model selection process to ensure that our model can most accurately reflect the spatial dependence and its dynamic impacts between geographic economic entities.

In the model selection process, we referred to the suggestions of Elhorst (2014) and Han and Xie (2017), and adopted a "specific to general" approach. First, we used the ordinary least squares method to perform regression analysis on the non-spatial effects model, and used the Lagrange multiplier test (LM) to determine the necessity of the spatial autoregressive (SAR) or spatial error (SEM) model. As you can see, after the preliminary LM-lag and LM-error tests, we further compared the R-LM-lag and R-LM-error according to the test results, and believed that the SDM model was the most suitable choice.

Furthermore, for the SDM model, we conducted likelihood ratio tests to determine whether there were fixed effects in time and space, and used the Hausman test to decide on the use of fixed effects or random effects estimation methods. In addition, we also applied the Wald test and LM test to determine whether the SDM model would degenerate into the SAR or SEM model. All these test results support the selection of the SDM model, indicating that it can comprehensively consider the effects of spatial autoregression and spatial error, and thus more fully capture and explain the spatial dependence in the research. All relevant test results and decision-making processes have been detailed in the corresponding sections of the paper, and the specific statistical outputs have been provided in Table 8. Furthermore, we have also included the regression results of SAR and SEM in Table 9.

Response to Reviewer 2:

Q1: Thank you very much for your valuable suggestions, pointing out the deficiencies in the research motivation of our paper. Based on your suggestions, we have comprehensively strengthened and expanded the research motivation part of the paper to more clearly demonstrate the importance and necessity of this research.

Q2: Thank you for your attention to our choice of mechanism variables. Regarding the variable of public environmental awareness you mentioned, we have previously analyzed its rationality in depth. The development of digital financial inclusion not only provides financing support for small and medium-sized enterprises, but also effectively improves the public's awareness and participation in environmental protection through digital platforms and technologies. These digital platforms use interactive educational content, information graphics, and other methods to popularize environmental protection knowledge and enhance public awareness of issues such as air pollution. Therefore, we believe that public environmental awareness is an important mechanism through which digital financial inclusion affects air quality.

Regarding the issue you raised that digital finance inclusion policies are mainly aimed at micro and small enterprises, and their impact on the general population may be relatively weak, we have carefully considered and analyzed the existing mechanism variable selections. We believe that although the policies are mainly aimed at small and micro enterprises, the resulting environmental impact has a spillover effect, indirectly affecting a wider range of people and the environment through changes in corporate behavior. Furthermore, our data and models have shown that these variables are statistically significant, thus demonstrating their rationality in explaining the relationship between digital financial inclusion and environmental pollution.

However, we deeply respect and value your opinion. If you believe that certain mechanism variables do not contribute much to understanding the research results or may cause misunderstandings, we are willing to consider removing these variables in the final draft. Please indicate whether such a modification is necessary, and we will follow your advice to further optimize our research.

Again, thank you for your detailed review and valuable suggestions.

Q3: Thank you for your attention to the innovation of our research. In this revision, we have clearly emphasized the important contributions and innovations made by this paper in understanding the interactive relationship between digital financial inclusion and environmental protection. Specifically:

1New theoretical perspective: We have explored the interaction between financial technology and environmental protection, not only analyzing the direct impact of digital financial inclusion on air pollution, but also deeply examining how it indirectly affects through improving public environmental awareness and promoting green technology innovation, which provides a new understanding of the relationship between digital finance and environmental protection.

2Spatial spillover effects and nonlinear characteristics: This paper reveals the spatial spillover effects and nonlinear characteristics of the positive effects of digital financial inclusion on air pollution improvement, demonstrating that its positive impacts can transcend geographical boundaries and exhibit different environmental effects at different stages of development. These findings have important implications for global environmental policies.

3Heterogeneity analysis: We have deeply explored the influence of geographical location and the strength of environmental regulations on the environmental effects of digital financial inclusion, revealing the role of economic development stages and regional characteristics in environmental protection. This detailed analysis is of great significance for formulating more refined and regionally adaptable environmental protection and financial policies.

4Analysis of policy incentive effects: Through empirical research, we have found that the "smart city" pilot policy and the "strengthen protection and avoid large-scale development" policy have a significant impact on the relationship between digital financial inclusion and environmental protection. This provides empirical support for policymakers to consider the synergistic effects of environmental protection and financial technology development.

In summary, this paper not only enriches the theoretical research on the relationship between digital financial inclusion and environmental protection, but also provides specific guidance for policymakers and practitioners on how to use digital financial tools to promote environmental protection, thereby establishing a bridge between theory and practice. We believe these innovations will help this research have a wide-ranging impact in both academic and practical application fields.

Q4: Thank you for the detailed review and valuable suggestions on Section 2.2. In response to the issues you pointed out regarding the lack of logical clarity and theoretical framework, we have comprehensively revised this section, clearly introducing environmental economics, social capital theory, etc. as the theoretical framework, and reorganized the content structure to ensure the logic and theoretical coherence of the discussion. In addition, we have also added corresponding literature references to strengthen the academic support for our arguments. We believe these improvements will significantly enhance the quality and persuasiveness of the paper.

Q5: Thank you for the thorough review of our paper and the suggestion to update the references. According to your instructions, we have checked and updated some of the references in the paper.

Q6: Thank you very much for your attention to our research and the valuable suggestions you have provided. We have added more relevant literature references to Section 2.3 based on your suggestion.

Q7: Thank you for your suggestion. Based on the research object and time span of this study, as well as the availability of data, we have decided to add two new control variables: infrastructure level (infra) and government intervention (gov).

Infrastructure level (infra): The inclusion of this variable aims to capture the potential impact of infrastructure development on environmental pollution. Good infrastructure, such as transportation, energy supply, and waste treatment facilities, can effectively reduce environmental pressure and improve air quality. The inclusion of this variable will help us more accurately explain the relationship between digital financial inclusion and environmental pollution, considering that infrastructure improvements may promote or inhibit certain pollution formation.

Government intervention (gov): Government policies and interventions play a crucial role in environmental management. Factors such as the intensity of environmental protection law enforcement, pollution control measures, and public investment all directly affect environmental quality. By including the government intervention variable, we can more comprehensively evaluate the actual effects of digital financial inclusion on environmental pollution in the policy context.

The addition of these control variables will help improve the explanatory power of our model and ensure the accuracy and reliability of the research results. Furthermore, this also makes our analysis more aligned with the real-world context, providing stronger support for policy formulation.

Q8: Thank you very much for your detailed review and valuable suggestions on our paper. After careful consideration, we have decided to remove Equation 2. We have also added more explanations for the relevant variables. Regarding the issue you mentioned about not being able to find relevant literature, we are very sorry. There was indeed an error in the spelling of the author's name in the original text, which may have caused difficulty in searching. Therefore, we now provide the correct literature citations and their DOIs for your and other readers' reference:

Hu, J., Yu, X. R., & Han, Y. M. (2023). Can ESG ratings promote corporate green transformation? Verification based on multi-time point DID method. Quantitative & Technical Economics Research, 40(7), 90-111. doi:10.13653/j.cnki.jqte.20230517.002.

Jiang, T. (2022). Mediating and moderating effects in empirical research of causal inference. China Industrial Economics, 40(5), 100-120. doi:10.19581/j.cnki.ciejournal.2022.05.005.

Q9: Thank you for your review of our research methods and the valuable suggestions you provided. Based on your comments, we have decided to remove the analysis part related to the Yangtze River Economic Belt from the heterogeneity analysis.

Q10: Thank you for pointing out the lack of analysis of Table 2 in our paper. We have further analyzed the descriptive statistics table.

Q11: Thank you very much for your review of our baseline regression results and the concerns you have raised. We have added more relevant literature support in the baseline regression section.

Q12: Thank you for pointing out our oversight regarding the labeling of the regression results. You correctly noted that we did not provide proper explanation for the values in parentheses in the regression results table, which may cause confusion among readers. The values in parentheses actually represent the standard errors of the estimated coefficients. Standard errors are an important statistical measure for evaluating the stability of estimated coefficients, as they reflect the degree of variation the coefficients may exhibit in repeated sampling. We have added an explanation of the standard errors in the note.

Q13: Thank you for the detailed review of the language quality of the paper. We have carefully considered your suggestions and made further proofreading and language refinement. Prior to the initial submission, this paper had undergone language editing through the Google Gemini service. To further improve the quality of the article, we have recently conducted another thorough proofreading.

Q14: Thank you for your suggestion. Following your guidance, we have further checked the paper and made modifications to the relevant errors.

---

## [Decision Letter · Decision Letter 1]

10 Jun 2024

The Impact of Digital Inclusive Finance on Environmental Pollution: A Case Study of Air Pollution

PONE-D-24-05850R1

Dear Dr. Fan,

We’re pleased to inform you that your manuscript has been judged scientifically suitable for publication and will be formally accepted for publication once it meets all outstanding technical requirements.

Kind regards,

Pengyu Chen

Academic Editor

PLOS ONE

Additional Editor Comments (optional):

Reviewers' comments:

Reviewer's Responses to Questions

**Comments to the Author**

1. If the authors have adequately addressed your comments raised in a previous round of review and you feel that this manuscript is now acceptable for publication, you may indicate that here to bypass the “Comments to the Author” section, enter your conflict of interest statement in the “Confidential to Editor” section, and submit your "Accept" recommendation.

Reviewer #1: All comments have been addressed

Reviewer #2: All comments have been addressed

2. Is the manuscript technically sound, and do the data support the conclusions?

Reviewer #1: Yes

Reviewer #2: Yes

3. Has the statistical analysis been performed appropriately and rigorously? 

Reviewer #1: Yes

Reviewer #2: Yes

4. Have the authors made all data underlying the findings in their manuscript fully available?

Reviewer #1: Yes

Reviewer #2: Yes

5. Is the manuscript presented in an intelligible fashion and written in standard English?

Reviewer #1: Yes

Reviewer #2: Yes

6. Review Comments to the Author

Reviewer #1: The author has carefully and correctly revised the opinions raised by two reviewers. I suggest that the manuscript meets the publication standards of PLOS ONE. Therefore, I offer an acceptance opinion.

Reviewer #2: The manuscript fits within the PLOS ONE scope of the study. Therefore, it is recommended for publication.

7. PLOS authors have the option to publish the peer review history of their article (what does this mean?). If published, this will include your full peer review and any attached files.

Reviewer #1: No

Reviewer #2: No

---

## [Editor Report · Acceptance letter]

18 Jun 2024

PONE-D-24-05850R1 

PLOS ONE

Dear Dr. Fan, 

I'm pleased to inform you that your manuscript has been deemed suitable for publication in PLOS ONE. Congratulations! Your manuscript is now being handed over to our production team.

Kind regards, 

on behalf of

Dr. Pengyu Chen 

Academic Editor

PLOS ONE